# Multifarious Translational Regulation during Replicative Aging in Yeast

**DOI:** 10.3390/jof8090938

**Published:** 2022-09-05

**Authors:** Tianyu Zhao, Asaka Chida, Yuichi Shichino, Dongwoo Choi, Masaki Mizunuma, Shintaro Iwasaki, Yoshikazu Ohya

**Affiliations:** 1Department of Integrated Biosciences, Graduate School of Frontier Sciences, The University of Tokyo, Kashiwa 277-8562, Japan; 2RNA Systems Biochemistry Laboratory, RIKEN Cluster for Pioneering Research, Wako, Saitama 351-0198, Japan; 3Unit of Biotechnology, Graduate School of Integrated Sciences for Life, Hiroshima University, 1-3-1 Kagamiyama, Higashi-Hiroshima 739-8530, Japan; 4Department of Computational Biology and Medical Sciences, Graduate School of Frontier Sciences, The University of Tokyo, Kashiwa 277-8561, Japan; 5Collaborative Research Institute for Innovative Microbiology, The University of Tokyo, Bunkyo-ku, Tokyo 113-8657, Japan

**Keywords:** ribosome profiling, replicative aging, *Saccharomyces cerevisiae*, translation

## Abstract

Protein synthesis is strictly regulated during replicative aging in yeast, but global translational regulation during replicative aging is poorly characterized. To conduct ribosome profiling during replicative aging, we collected a large number of dividing aged cells using a miniature chemostat aging device. Translational efficiency, defined as the number of ribosome footprints normalized to transcript abundance, was compared between young and aged cells for each gene. We identified more than 700 genes with changes greater than twofold during replicative aging. Increased translational efficiency was observed in genes involved in DNA repair and chromosome organization. Decreased translational efficiency was observed in genes encoding ribosome components, transposon Ty1 and Ty2 genes, transcription factor *HAC1* gene associated with the unfolded protein response, genes involved in cell wall synthesis and assembly, and ammonium permease genes. Our results provide a global view of translational regulation during replicative aging, in which the pathways involved in various cell functions are translationally regulated and cause diverse phenotypic changes.

## 1. Introduction

Aging is accompanied by noticeable diverse phenotypic changes, which can be regarded as a natural process of the life cycle. Aging is also a factor associated with various fatal diseases. As aging occurs universally in vertebrates, anti-aging research has been conducted using vertebrates to avoid health complications [1,2].

Extensive aging-focused research has been conducted using several model organisms other than vertebrates [3]. Model organisms suitable for aging research include experimental organisms such as the budding yeast *Saccharomyces cerevisiae*, nematode *Caenorhabditis elegans*, and fruit fly *Drosophila melanogaster*. These organisms all have short lifespans, are economically feasible, and have abundant established genetic tools [4]. Using a genetic approach with these invertebrate model organisms, hundreds of genes have been identified that lead to lifespan extension when mutated [5]. Compounds that affect lifespan have been extensively analyzed, thus providing insights into the extension of human health and longevity [6,7].

*S. cerevisiae* has the shortest generation time among the model organisms commonly used for aging research. Several age-dependent hallmarks have been reported in this organism, including mitochondria dysfunction [8,9], transcriptional regulation [10], nuclear and mitochondrial genetic instability [11,12], change in telomere length [13], asymmetric cell division [14], and characteristic metabolic fluxes [15]. Dietary restriction increases the lifespan of yeast, like other organisms [16,17,18]. Studies of the yeast replicative lifespan (RLS), defined as the number of cell divisions before senescence, required micromanipulation to separate mother and daughter cells at each division, entailing considerable effort [19,20]. The use of micromanipulation also made it difficult to apply omics approaches, requiring large amounts of materials. Recently, new methods have been developed, including the mother enrichment program [21], column-based cultivation method [22], and miniature chemostat aging device (MAD) [23]; unlike microfluidic platform systems, these can collect many cells [24]. These methods have supported omics studies of the metabolome [25], epigenome [26], transcriptome via RNA sequencing (RNA-Seq) [23,27], and proteome [15]. These studies have gradually elucidated the comprehensive dynamic behavior of intracellular chemistry during replicative aging.

The correlation between mRNA and protein levels evidently becomes progressively decoupled during replicative aging [22], suggesting that translation changes in aged cells. Previous studies have indicated that bulk protein synthesis is significantly reduced during aging in a wide range of organisms and cell types [28], including yeast cells [29,30]. However, the comprehensive landscape of translational regulation has not been fully explored. Here, utilizing ribosome profiling, we showed that aged yeast cells reprogram their translation profiles. We successfully collected numerous dividing aged cells under unstressed conditions with the MAD system, in which mother cells are trapped in a vessel using a magnet and cultured using chemostat technology. We observed translation reduction in genes involved in translation, transposons, stress response, cell wall synthesis, and ammonium uptake. Our results shed light on the mechanism of translation modulation during replicative aging.

## 2. Materials and Methods

### 2.1. Assembly of MADs

The MAD system was assembled as described by Hendrickson et al. [23], with some modifications (Appendix A). Peristaltic pumps (Eyela, Cat. # MP-2000 and Cat. # MP-2100, Tokyo RIKAKIKAI Co., Ltd., Tokyo, Japan) were used to control media flow. An air pump (Gex, Cat. # 2000SB, GEX. Co., Ltd., Osaka, Japan) and air pressure gauge (Daiichi Keiki, Cat. # EA729DW-10, DAIICHI KEIKI SEISAKUSHO Co., Ltd., Hyogo, Japan) were used to control the aeration of each ministat vessel. Tube connection was simplified: no blunt-end needle was needed, and a silicone tube was used instead of the Marprene tube. Neodymium ring magnets (ABM Magnetics, Cat. # RY0X04, ABM Magnetics Co., Ltd., Shehzhen, China) and Dynamag-50 (Thermo Fisher Scientific, Cat. # 12302D, Thermo Fisher Scientific Inc., Waltham, MA, USA) were used for bead-based purification of aged cells.

### 2.2. Preparation of Aged Yeast Cells

The diploid wild-type yeast strain BY4743 (*MATa/MATα his3Δ1/his3Δ1 leu2Δ0/leu2Δ0 LYS2/lys2Δ0 met15Δ0/MET15 ura3Δ0/ura3Δ0*) was grown in synthetic defined (SD) medium containing 0.17% yeast nitrogen base without amino acids or ammonium sulfate (BD Biosciences, Franklin Lakes, NJ, USA), 0.5% ammonium sulfate (Wako Chemicals, Osaka, Japan), 0.5% casamino acids (BD Biosciences, Franklin Lakes, NJ, USA), 2% glucose (Wako Chemicals), and 2% D (+)-mannose (Wako Chemicals) overnight at 25 °C with shaking at 180–200 RPM and harvested at 5 × 10^6^ cells/mL (early log phase). For each MAD, 10^8^ yeast cells were washed twice in 0.6 mL phosphate-buffered saline (PBS) at 2000× *g* for 5 min at room temperature. During the second wash, EZ-Link Sulfo-NHS-LC-LC-Biotin reagent (Thermo Fisher Scientific, Cat. # 21338) was weighed out (2.0 mg) and suspended in 0.4 mL PBS. The washed cell pellets were resuspended in 0.6 mL PBS and combined with EZ-Link Sulfo-NHS-LC-LC-Biotin in PBS at a final reaction volume of 1 mL. The labeling reaction was conducted for 30 min at room temperature with rotation followed by two washes in PBS, as described above. Biotin-labeled cells were resuspended in a SD medium and batch-cultured in flasks at 25 °C with shaking at 180–200 RPM for 13 h.

Grown cells were filtered using the Stericup-GP Quick Release Sterile Vacuum Filtration System (Millipore, Cat. # S2GPU05RE, Merck, Daemstadt, Germany), suspended in 40 mL PBS, washed once with PBS, and collected in a 50 mL Falcon centrifuge tube. Beading reactions were then conducted by combining the resuspended cells with washed magnetic streptavidin beads (Thermo Fisher Scientific, Dynabeads MyOne Streptavidin C1, Cat. # 65001) and rotating the mixture at room temperature for 30 min. The beaded mother cells were collected by placing the beading reaction on a magnet (Thermo Fisher Scientific, Dynamag-50 Magnet Cat. # 12302D) for 10 min. The supernatant fraction containing daughter cells was carefully removed and collected as two biological replicates of young cells (Young, Figure 1A). For operation of the MAD, the supernatant was loaded twice onto the magnet to collect all beaded mother cells, which were then re-suspended in a SD medium (final volume of 1.0 mL). Before this suspension was loaded into the MAD, the beaded mother cells were collected for analysis as two biological replicates of middle-aged cells (Middle, Figure 1A).

MAD loading and MAD operation were performed in accordance with the methods of Hendrickson et al. [23]. The loaded ministat was placed on a neodymium ring magnet, and cells were allowed to bind for 10 min before resumption of pumping (50 rpm; flow rate ~25 mL/h) and air flow (25–33% of max output; approximately 1 PSI). The loaded cells were then grown in MAD for 24 h at 25 °C; beaded cells were collected in SD medium and placed back on the magnet for 5–10 min. This washing step was repeated twice. The cleaned beaded cells were resuspended in 1.0 mL SD medium and collected as aged cells (Aged, Figure 1A). The purity and viability of aged cells were > 80% and > 95%, respectively.

### 2.3. Cell Harvest

Yeast cells were collected on 0.45 μm pore size membrane filters (Millipore, Cat. # HAWP04700) via filtration, immediately removed with a cell scraper (TPP, Cat. # 99002, TPP Techno Plastic Products AG, Trasadingen, Schaffhausen, Switzerland), then placed into Nunc 50 mL conical sterile polypropylene centrifuge tubes (Thermo Fisher Scientific, cat. No. 339652) containing liquid nitrogen. Then, 0.6 mL lysis buffer (20 mM Tris-HCl pH 7.5 (Wako Pure Chemical Industries, Ltd., Cat. # 318-90225, Wako Pure Chemical Industries, Ltd., Osaka, Japan), 150 mM NaCl (Nacalai Tesque, Cat. # 06900-14, NACALAI TESQUE, Inc., Kyoto, Japan), 5 mM MgCl2 (Nacalai Tesque, Cat. # 20942-34), 1 mM dithiothreitol (New England Biolabs, Cat. # M0368L, New England Biolabs Inc., Ipswich, MA, USA), 100 µg/mL cycloheximide (Sigma-Aldrich, Cat. # C4859-1ML), 100 µg/mL chloramphenicol (Wako Pure Chemical Industries, Cat. # 030-19452), and 1% Triton X-100 (Nacalai Tesque, Cat. # 12967-32)) was added dropwise into the tubes to obtain flash-frozen droplets. Finally, the tubes were stored at −80 °C until the liquid nitrogen evaporated.

### 2.4. Cell Lysis

The frozen samples were individually transferred to 3 mL cryotubes (Yasui Kikai, Cat. # ST-0320PCF, Yasui Kikai Corporation, Osaka, Japan) along with a metal bead (Yasui Kikai, Cat. # MC0316 (S)) in a Styrofoam box containing liquid nitrogen. Then, cell lysis was performed using a Multi-beads Shocker (Yasui Kikai, Cat. # MB2200 (S)) at 2800 RPM for 15 s. After melting on ice, the sample was transferred to a 1.5 mL tube and treated with 15 U Turbo DNase (Thermo Fisher Scientific, Cat. # AM2238) on ice for 10 min. The lysate was cleared by centrifugation at 20,000× *g* for 10 min at 4 °C. The RNA concentration of lysate was measured using the Qubit RNA BR Assay Kit (Thermo Fisher Scientific, Cat. # Q10210).

### 2.5. Library Preparation for Ribosome Profiling and RNA-Seq

The ribosome profiling library was prepared in accordance with previously described protocols [31,32]. Briefly, 300 μL cell lysate containing 20 µg RNA was incubated with 10 U RNase I (Epicentre, Cat. # N6901K, Epicentre, Paris, France) at 25 °C for 45 min. Ribosome-protected RNA fragments of 17–34 nt were excised from the gel after polyacrylamide gel electrophoresis. The Ribominus Transcriptome Isolation Kit, yeast (Invitrogen, Cat. # K1550-03, Thermo Fisher Scientific Inc., Waltham, MA, USA) was used for rRNA depletion.

For RNA-Seq, RNA was extracted from the same lysate used for ribosome profiling with TRIzol LS reagent (Thermo Fisher Scientific, Cat. # 10296-010) and a Direct-zol RNA Microprep Kit (Zymo Research, Cat. # R2060, Zymo Research Corporation, Irvine, CA, USA). Then, 0.5 µg RNA was subjected to rRNA depletion using the Ribominus Transcriptome Isolation Kit, followed by library preparation using the TruSeq Stranded mRNA Library Prep Kit (Illumina, San Diego, CA, USA).

Library sequencing, short-read sequence analysis (2 × 150 bp) using a HiSeq X sequencer (Illumina), was performed by Macrogen Japan (Tokyo, Japan).

### 2.6. Data Analysis

Sequence data were processed as previously described [31], with the following modifications. Read quality filtering and adapter trimming (read 1 for ribosome profiling and read 1/2 for RNA-Seq) were performed with Fastp [33]. After non-coding RNA-mapped reads had been removed, the remaining reads were aligned to the budding yeast genome S288C (R64, sacCer3) using STAR 2.7.0a [34]. For ribosome profiling, the A-site offsets were determined as 13 for 26 nt, 14 for 27 nt, 15 for 21–22 nt and 28–29 nt, and 16 for 30–31 nt footprints based on the location of the peaks of 5′ ends of footprints along the coding sequence (CDS), which represents the ribosome on the start codon. For RNA-Seq, an offset of 15 was used for all mRNA fragments. The fold changes of read counts were calculated using the DESeq2 package [35]. Reads corresponding to the first and last five codons of the CDS were excluded from calculation. Translational efficiency was defined as over- or under-representation of ribosome profiling counts by RNA-Seq counts and calculated using DESeq2 with a generalized linear model. All GO enrichment analyses [36,37] were performed online using the GO Term Finder [38] tool (https://www.yeastgenome.org/goTermFinder) (accessed on 15 December 2021). Reads containing the intron of the *HAC1* gene were counted using Sashimi plot tools in the Integrative Genome Viewer [39,40].

To map the reads to Ty genes, we extracted sequences for the CDSs and 300 nt upstream and downstream of all Ty1–4 genes from the *Saccharomyces* Genome Database (SGD) and obtained consensus sequences using MAFFT [41,42] ver. 7 (https://mafft.cbrc.jp/alignment/server/) (accessed on 10 April 2022) with the default settings. Then, the reads remaining after removing non-coding RNA were aligned to the consensus sequences as described above. The overlapped regions between *TyA* and *TyB* were excluded from the analyses.

All custom scripts used in this study are available upon request.

### 2.7. Biochemistry

To examine Fks1 levels, whole-cell protein extracts were prepared as described previously [43]. Briefly, cells were pelleted, treated with a 1 mL N/β solution (0.25 N NaOH, 1% β-mercaptoethanol), and incubated on ice for 10 min. Then, 100 μL of 100% trichloroacetic acid was added, and incubated on ice for 10 min. The supernatant was discarded after centrifugation at 15,000× *g* for 2 min at 4 °C. After the pellet was washed with 500 μL of 1 M Tris-HCl (pH 8.0), the cells were resuspended in sodium dodecyl sulfate (SDS) sample buffer including 62.5 mM Tris-HCl (pH 6.8), glycerol (10%), SDS (2%), β-mercaptoethanol (2%), and bromophenol blue (0.005%), incubated for 10 min at 37 °C, and pelleted. Next, the supernatants were loaded in a mini-gel (8%; Bio-Rad, Bio-Rad Laboratories, Inc., Hercules, CA, USA), and Western blotting was performed with mouse monoclonal antibody to Fksl (T2B8) [44]. Horseradish peroxidase-conjugated secondary antibody was obtained from Promega (Cat. # W4021) and proteins were detected with an enhanced chemiluminescence system (Amersham, ECL Plus). Grayscale analysis was performed using ImageJ.

### 2.8. Fluorescence Microscopy

Bud scars were stained with wheat germ agglutinin conjugated to Alexa Fluor 488 (Thermo Fisher Cat. # W11261) for 30 min, then observed under a fluorescence microscope (Axio Imager M1, Zeiss, Jena, Germany).

### 2.9. Database

The databases used in this study are SGD (https://www.yeastgenome.org) (accessed on 15 December 2021), PAXdb: Protein Abundance Database (https://pax-db.org) (accessed on 12 February 2022), and uORFlight (http://uorflight.whu.edu.cn) (accessed on 16 February 2022).

## 3. Results and Discussion

### 3.1. Outline of the Experiments

To investigate global translation dynamics during replicative aging, we applied ribosome profiling and RNA-Seq to yeast cells cultured in the MAD system [23], in which mother cells are trapped in a vessel surrounded by custom-sized neodymium ring magnets (Figure 1A). First, we labeled the cell wall of exponentially grown cells with biotin and grew them for 13 h in liquid culture. Then, biotinylated mother cells were captured with magnetic streptavidin beads and collected as middle-aged cells (Middle, Figure 1A). As daughters produced from the mother cells did not inherit the biotin coat and magnetic beads, cells that were not coated with magnetic beads were collected as young cells in the flow-through fraction (Young, Figure 1A). The labeled mother cells were loaded into the MAD, then further incubated for 24 h to obtain aged cells (Aged, Figure 1A). Young, Middle, and Aged cells had divided approximately 0–2 times, 10 times, and 20 times, respectively (Figure 1B). The translation process is affected by various stresses such as temperature shifts, medium changes, and physical damage. Therefore, the experiments were conducted using a slightly modified MAD protocol to minimize the stress caused by experimental operations (Figure 1A). The numbers of centrifuging steps and PBS washes were kept consistent among conditions. In the final cell harvesting step, filtration was used, which had only a mild effect on the yeast cells.

### 3.2. Validation of Our Samples with RNA-Seq Data

To validate our aged yeast cell samples, we next tested whether we could reproduce the aging hallmarks previously reported in RNA-Seq data. We conducted Gene Ontology [36,37] (GO) enrichment analysis of significantly upregulated genes in Middle cells with a threshold of *p* < 0.05 to identify important gene functions during aging. Upregulated genes were significantly enriched in mitochondrion-related GO processes, including the mitochondrion (GO: 0005739) with 2.2-fold enrichment; mitochondrial electron transport, ubiquinol to cytochrome c (GO: 0006122) with 18.6-fold enrichment; and mitochondrial electron transport, cytochrome c to oxygen (GO: 0006123) with 15.3-fold enrichment (Appendix A). The oxidation-reduction process (GO: 0055114) was also significantly enriched, showing 4.3-fold enrichment (*p* = 9.67 × 10^−22^) (Appendix A), as previously reported. Thus, our RNA-Seq analysis confirmed the important role of mitochondria in aging, validating our aging samples.

### 3.3. Quality Control of Ribosome-Protected mRNA Footprints

The quality of ribosome-protected mRNA footprints was assessed via metagene analysis. First, the length distribution of the ribosome footprints obtained from Young, Middle, and Aged cells was investigated (Appendix A). The largest peak representing ribosomes accommodating A-site tRNA, and the second largest peak representing ones with no tRNA at the A-site, appeared around 27–28 nt and around 20–21 nt, respectively; these results were indicative of ribosome profiling success for all samples [45]. Then, the 5′ end of the accumulating ribosome footprints obtained from Young, Middle, and Aged cells was examined (Appendix A). We observed that the ribosome footprint sequences matched the protein-coding portion of the transcript, extending from the position −12 nt from the start that corresponded to the AUG initiation codon of the P-site. In addition, the footprint-allocated position of the A-site showed a strong preference for the first nucleotide position of each codon, consistent with the reading frame; it exhibited 3-nt periodicity. Third, the protein-coding boundary and 3-nt periodicity were observed at the 5′ end of the accumulating ribosome footprints around stop codons obtained from Young, Middle, and Aged cells (Appendix A). These results indicate that even samples from Aged cells guarantee the quality of the ribosome-protected mRNA footprints.

### 3.4. Translation Profile Changes during Replicative Aging

Ribosome profiling and RNA-Seq data of Young and Middle cells enables us to investigate changes in global gene expression during replicative aging. We observed a broad spectrum of gene expression changes during aging, at both transcriptional and translational levels (Figure 2A). For most genes with changed expression, regulation occurs at the transcriptional level without changes in translational efficiency, that is, the ratio of ribosome footprint counts to total RNA counts of each gene remains the same. The fold change of ribosome footprints is almost equal to that of transcripts with a correlation coefficient of 0.77 (Figure 2A). However, we also detected genes with greater than twofold upregulation and twofold downregulation translationally with age: overall, 133 and 233 genes had upregulation and downregulation by more than twofold, respectively (Figure 2B, Appendix A). We found that genes with greater than twofold increases in translational efficiency with age were significantly enriched (*p* < 0.05) in the processes of chromosome organization (GO: 0051276) and DNA repair (GO: 0006281) (Figure 3A, Appendix A). On the other hand, more GO categories were enriched among the downregulated genes (*p* < 0.05), including cytoplasmic translation (GO: 0002181), amide biogenesis (GO: 0043604), organonitrogen compound biogenesis (GO: 1901566), and cellular amide metabolism (GO: 0043603) (Figure 3A, Appendix A). Markedly altered genes are listed in Appendix A. Among the 10 downregulated genes, 80% were retrotransposon-related genes; the remaining genes were *CCW12* and *PMA2*.

The tendency for more genes to be downregulated was stronger in Aged cells. In total, 221 and 504 genes exhibited greater than twofold increases and decreases in translational efficiency, respectively (Figure 2B,C, Appendix A). The upregulated genes were significantly enriched (*p* < 0.05) in the processes of chromosome segregation (GO: 0007059) and mitotic sister chromatid segregation (GO: 0000070) (Figure 3B, Appendix A). The downregulated genes were significantly enriched (*p* < 0.05) in the processes of ribosome biogenesis (GO: 0042254), maturation of SSU-rRNA (GO: 0030490), rRNA processing (GO: 0006364), and ncRNA processing (GO: 0034470) (Figure 3B, Appendix A). Among the downregulated genes, 23 were markedly downregulated (Appendix A), 87% of which were retrotransposon-related genes; the remaining genes were *HAC1*, *MEP3*, and *SYG1*.

### 3.5. Translation of Ribosome Components Decreases during Aging

Genes with decreased translational efficiency with age were significantly enriched in the processes of cytoplasmic translation, suggesting global downregulation of ribosome components during aging. Therefore, we investigated translational efficiency, RNA-Seq, and ribosome footprints of ribosome components in Middle and Aged cells. The yeast ribosome consists of small (40S) and large (60S) subunits, which are composed of 79 proteins in addition to rRNA. The translational efficiency of many ribosome components decreased more than twofold: the translational efficiency of 22 and 9 components decreased in Middle and Aged cells, respectively (Figure 4A,B). As most of the transcriptional activity of ribosome components was reduced, ribosome footprints also decreased in the Middle cells (Figure 4C). Intrinsic large subunit components such as L15, L20, and L41 were downregulated [46]. In comparison, the transcriptional activity of ribosome components increased overall in Aged cells, thus ribosome footprints increased (Figure 4D). From these results, we conclude that ribosome components are downregulated translationally during the aging process, while their transcription is initially downregulated and then later upregulated.

### 3.6. Translation of Ty1 and Ty2 Transposon Genes Decreases during Aging

We noticed that most genes that were markedly downregulated translationally were retrotransposon-related genes (Appendix A). Budding yeast retrotransposons include five families; Ty1, Ty2, Ty4, and Ty5 are considered copia-type (Pseudoviridae), while Ty3 is considered gypsy-type (Metaviridae) [47,48]. In our laboratory strain, S288C, the most abundant transposons are Ty1 (31 copies) and Ty2 (13 copies). These closely related 5.9 kb full-length elements consist of two overlapping open reading frames (ORFs): *TyA* and *TyB* [49]. As the sequence of each Ty family is quite similar, we remapped the RNA-Seq and ribosome profiling reads to the consensus sequences of Ty1–Ty4 to obtain more reliable results. We first confirmed that the transcriptional levels of Ty1 increased during aging (Figure 5A), as reported previously [50]. In particular, its transcriptional level increased more than twofold in Aged cells. On the other hand, the transcriptional levels of other Ty families decreased (Figure 5A). In addition, we found that translational efficiency decreased more than twofold in Ty1 and Ty2 in Middle and Aged cells (Figure 5B). As a result, their final ribosome footprints were reduced (Figure 5C). As the +1 frameshifts of *TyA* and *TyB* ORFs were observed properly (Appendix A), we assumed that the ribosome footprints from Ty were detected correctly. These results suggest that downregulation at the translation level occurs particularly in Ty1 and Ty2, possibly making it difficult for Ty translocation to occur.

We found that genes involved in DNA repair pathways and chromosomal organization including *RAD53* and *RAD54*, which encode a DNA damage response kinase and a stimulation factor for DNA strand exchange, respectively, were upregulated. Given that Ty1 gene expression increases during replicative aging [50], the aged cells may activate the yeast DNA repair pathway due to the frequent occurrence of Ty1 integration. However, this is unlikely, because we observed ribosome footprint reductions of Ty1 and Ty2 retrotransposon in aged cells. Therefore, DNA damage response pathways may be activated by mechanisms other than transposition of retrotransposons in aged cells. As the active expression of mitochondrial-related genes in Middle cells reflected a high burden on mitochondrial function, we propose that the DNA damage response may be caused by oxidative reaction with reactive oxygen species (ROS), which is one of the main causes of endogenous DNA damage [51]. We also found evidence suggesting that the aged cells suffered from ROS stress; the transcriptions of superoxide dismutase encoded by *SOD1* and *SOD2*, and catalase encoded by *CAT2*, *CAT8*, and *CTT1* were upregulated (Appendix A).

### 3.7. Translational Reduction of HAC1 Leads to Transcriptional Repression of Genes Involved in Protein Folding

The translational efficiency of *HAC1* decreased to ≤1/8 of its initial value in Aged cells (Figure 2C, Appendix A). Hac1, a basic leucine zipper transcription factor, is regulated in response to endoplasmic reticulum (ER) stress and promotes the transcriptional activation of downstream genes involved in the unfolded protein response (UPR) [52,53]. This suggests that the *HAC1*-dependent UPR, which occurs constitutively without extrinsic stress, is perturbed during replicative aging. To confirm this, we examined the expression levels of representative downstream genes of *HAC1*, including *KAR2*, *FPR2*, *PDI1*, *LHS1*, *EUG1*, and *SCJ1* [54,55,56]. RNA-Seq analysis revealed that five of the six downstream genes showed greater than twofold reductions (false discovery rate of *KAR2* = 2.61× 10^−7^, *LHS1* = 3.04 × 10^−6^, and *SCJ1* = 4.40 × 10^−4^) in gene expression in Aged cells (Figure 6A,B). A fivefold decrease in transcription level was observed for *KAR2*, which is involved in several processes related to ER functions. No significant change in *HAC1* gene expression was observed. Although translational control of *HAC1* usually occurs through Ire1-mediated mRNA splicing [54,55,56], this control mechanism seemed unlikely during aging, because we also observed the reduction of intron-containing reads in Aged cells (Appendix A). These results suggest that the decreased translation in *HAC1* may attenuate the UPR during replicative aging.

Quality control of proteins in the ER is perturbed during replicative aging [57]. Previous studies indicated that *hac1∆* [58] and an ER-stress response mutant, *med2∆*, have a short RLS [59]. In this study, we observed decreases in *HAC1* translation and RNA levels of downstream genes during replicative aging. This has a negative effect on RLS. Translational control of *HAC1* occurs through Ire1-mediated mRNA splicing [54,55,56]. However, we noted a reduction in intron-containing reads in Aged cells, implying that translation reduction of *HAC1* may be regulated independently from the splicing-mediated mechanism.

### 3.8. Decreased Cell Wall Biosynthesis during Replicative Aging

The translational efficiency of the mannoprotein encoding gene *CCW12* decreased by 6.3-fold during replicative aging (Figure 2A, Appendix A). The yeast cell wall is composed of non-filamentous mannoproteins and filamentous polysaccharides (1,3-β-glucan, 1,6-β-glucan, and chitin), which form a firm extracellular matrix like reinforced concrete [60]. Although no genes encoding mannoproteins are essential to fitness during cell proliferation, they have pivotal roles in the establishment of cell shape. The most important mannoprotein is Ccw12 [61]; the *CCW12* gene encodes a mannoprotein that constitutes 40% of total mannoprotein in a cell, plays a major structural role, and contributes strongly to cell morphogenesis. Examination of changes in translation of other major mannoproteins revealed that mannoprotein genes such as *CWP2*, *ECM33*, *TIP1*, and *CCW14* decreased translationally by more than twofold (Figure 7A). There were no significant changes in the RNA levels of these genes (Figure 7B). The other minor mannoprotein genes (e.g., *CWP1*, *DAN4*, *PST1*, and *TIR1*) were also downregulated (Appendix A). In addition, we found that *FKS1*, encoding the catalytic subunit of 1,3-β-glucan synthase, decreased transcriptionally and translationally by more than twofold (Figure 7C,D). Western blotting experiments also indicated the reduction of Fks1 protein level in Middle and Aged cells (Figure 7E,F). Thus, deficiencies in mannoprotein and 1,3-β-glucan biosynthesis may lead to cell wall instability during replicative aging.

Our data suggest translational regulation of genes that affect cell wall synthesis during replicative aging. Therefore, cell wall synthesis is presumably insufficient in aged cells, causing the cell wall to become fragile. This insufficiency may explain why some cells burst at the end of their RLSs [62].

### 3.9. Decreased Ammonium Permease during Replicative Aging

For ammonium uptake and sensing, budding yeast requires ammonium permease localized to the plasma membrane, which is encoded by *MEP1*, *MEP2*, and *MEP3* [63,64,65]. As one ammonium permease, *MEP3*, showed marked downregulation of translational efficiency in Aged cells (*p* < 0.01, Figure 2C, Appendix A), we examined the translational changes of the other two ammonium permeases. As shown in Figure 8A, the translation of *MEP1* and *MEP2* also decreased by more than twofold. There were no significant changes in the expression of these genes (Figure 8B). These results suggest that aged yeast cells have decreased ammonium permease levels.

During NH_4_^+^ uptake, intracellular charge balance is maintained by plasma membrane H^+^-ATPase [64], which is encoded by *PMA1* and *PMA2* [66]. We found that the translational efficiency of the minor gene *PMA2* was decreased by more than eightfold in Middle cells. Although the major gene *PMA1* showed no marked translational downregulation (Figure 8C), it showed decreased gene expression level (Figure 8D, Appendix A). These results suggest that *PMA1* and *PMA2* are differentially downregulated at the transcriptional and translational levels, respectively.

The most strongly downregulated *MEP3* gene contains an upstream ORF (uORF), which is present in only 8.3% of normal yeast genes [67]. Since uORFs are involved in the translational control of *GCN4* and *CPA1* [68], it is possible that uORFs affect the translational efficiency in aged cells. However, we have no direct evidence that the uORF in *MEP3* plays a role in translational repression. The mechanism of translational regulation of *MEP3* during replicative aging warrants further research.

### 3.10. Comparison with Other Aging Cells

Translational efficiency has been found to be downregulated in genes during chronological aging, including ribosome genes and genes involved in cytoplasmic translation [69]. However, it has also been reported to be associated with the upregulation of genes during chronological aging, including genes related to autophagy, ion homeostasis, and aerobic respiration [69]. These divergent findings may reflect the essential difference between the RLS and chronological lifespan in yeast. Studies of aged mouse liver, spleen, and muscle cells also showed that genes involved in cytoplasmic translation were downregulated with increasing translational efficiency [70]. Contrasting our results, genes involved in the tricarboxylic acid cycle and oxidative phosphorylation are affected in aged mouse cells [30,70]. Genes related to the extracellular matrix were upregulated during aging in mice, but downregulated during aging in yeast. Taken together, the genes that are upregulated in association with translational efficiency during aging may have implications for characterization of cellular aging.

We propose a new mechanism for translational regulation in replicative aging through ribosome profiling and RNA-seq of aged cells isolated by the MAD system. Diverse cellular processes including translation, transposons, UPR, cell wall synthesis, and ammonium uptake were translationally downregulated in aged cells (Figure 9), although the translational efficiency of genes involved in chromosomal organization and DNA repair pathways were increased. A previous translatome study that used the mother enrichment program, which is an inducible genetic system that eliminates daughter cells specifically, has reported only a global decline in the translation process, presumably because the employed method confers stress to aged cells and results in minimal cell division [71]. The MAD system used in this study enabled the collection of dividing aged cells under the least stressful conditions.

## 4. Conclusions

More genes exhibited translational downregulation than upregulation in dividing aged yeast cells. Among the small number of upregulated genes are genes involved in DNA repair and chromosome organization. The numerous downregulated genes included those involved in cytoplasmic translation, *TyA* and *TyB* genes of transposons Ty1 and Ty2, the transcription factor *HAC1* that functions in the UPR, genes involved in cell wall synthesis and assembly, and ammonium permeases. We propose that the pathways involved in various cell functions are predominantly regulated translationally, linking diverse phenotypic changes that occur during aging.

## Figures and Tables

**Figure 1 jof-08-00938-f001:**
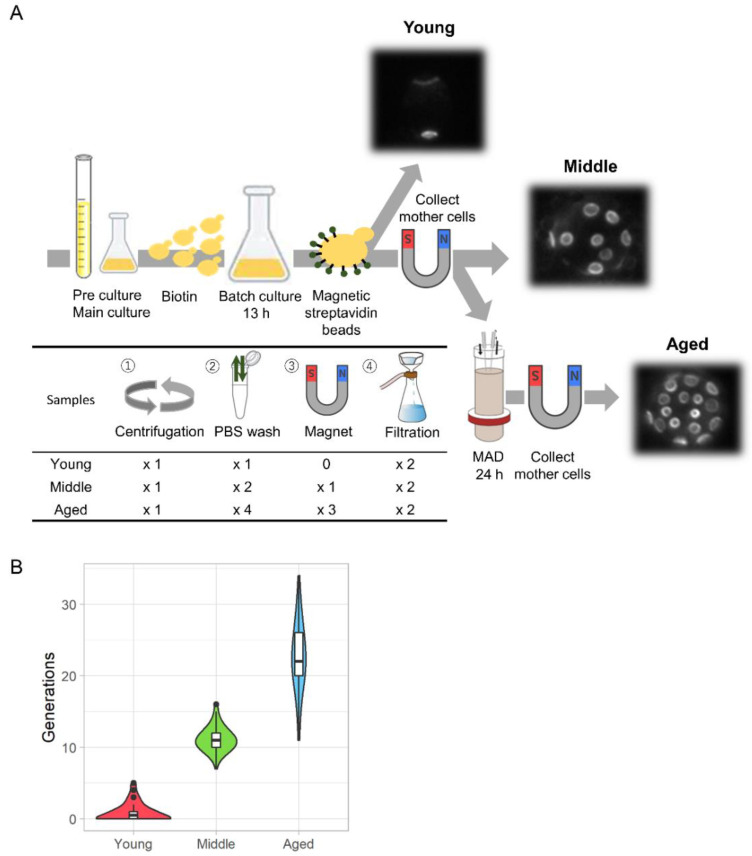
Outline of the experiments. (**A**) Young, Middle, and Aged cells were collected from MAD for RNA-Seq and ribosome profiling experiments. Grey arrows indicate workflows. Yeast cells were treated with EZ-Link Sulfo-NHS-LC-LC-Biotin, collected, and batch-cultured in flasks at 25 °C for 13 h. Middle-aged mother cells were bound to magnetic streptavidin beads, then collected with a magnet (Middle). Supernatant containing the daughter cell fraction was also collected (Young). MAD loading and MAD operation were performed in the vessel associated with the magnet and incubated for an additional 24 h. Finally, the beaded aged cells were collected and purified with a magnet (Aged). The numbers of centrifugations, PBS washes, magnet operations, and filtrations are shown in the inset table. Efforts were made to minimize differences in conditions among samples. (**B**) Distribution of mother cell ages estimated from the numbers of bud scars. Bud scars were counted via staining with wheat germ agglutinin conjugated to Alexa Fluor 488 for 30 min, then observation under a fluorescence microscope.

**Figure 2 jof-08-00938-f002:**
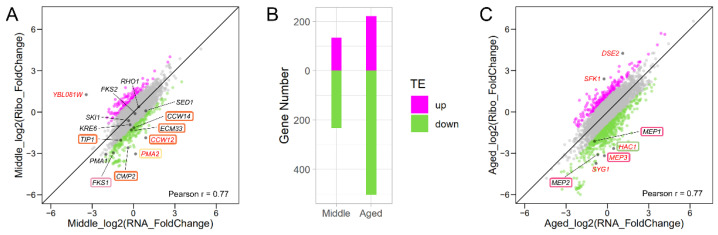
Global analysis of translational efficiency changes in Middle and Aged yeast cells. (**A**) Translational efficiency changes in Middle cells. Genes with translational efficiency increases of more than twofold (magenta), no notable changes (grey), and decreases of more than twofold (green) are plotted. Genes with markedly altered translational efficiency are shown in black with red gene name. Genes with markedly altered translational efficiency are shown in colorful boxes, same color means in a same group. Young yeast cells were used as the control. (**B**) Genes with upregulated and downregulated translational efficiency (TE) in Middle and Aged cells are shown. (**C**) Translational efficiency changes in Aged cells. Color legend is identical to (**A**).

**Figure 3 jof-08-00938-f003:**
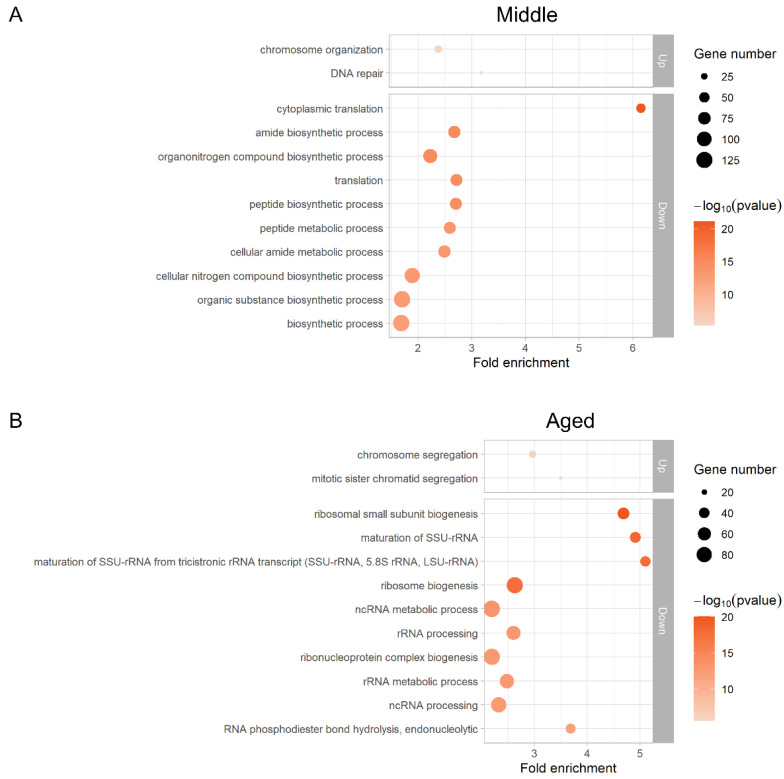
GO enrichment results of genes with significantly changed translational efficiency. Upregulated genes with changes of more than twofold compared to Young cells were analyzed (*p* < 0.05). Fold enrichment and *p*-values are shown. The size of the circle indicates the number of genes associated with the GO term. (**A**) Genes with greater than twofold changes in Middle cells. (**B**) Genes with greater than twofold changes in Aged cells.

**Figure 4 jof-08-00938-f004:**
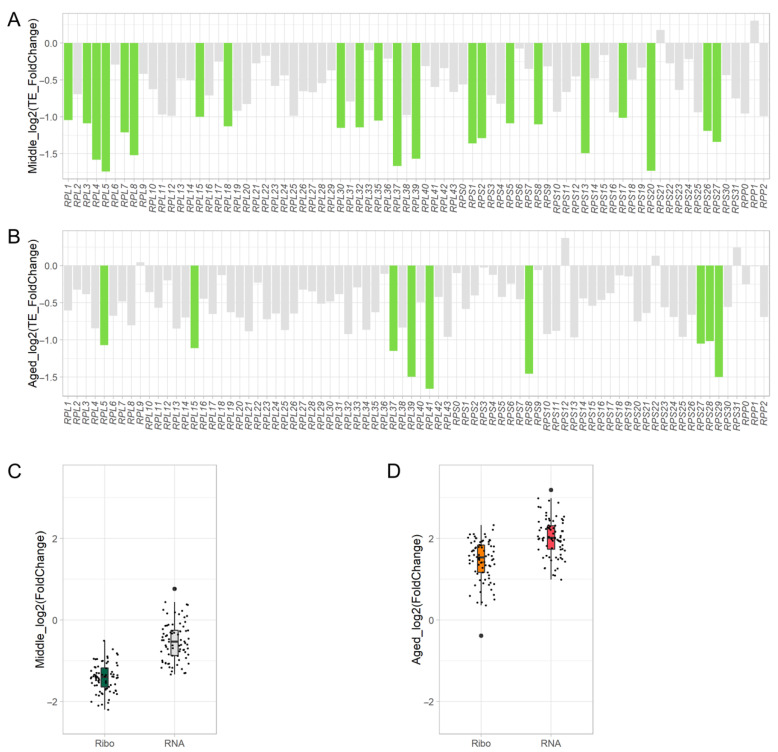
Translation of ribosome components decreases during aging. Changes in translational efficiency (TE) of ribosome proteins in Middle (**A**) and Aged (**B**) cells. Green indicates downregulation by more than twofold. Young yeast cells were used as the control. Changes in Ribo-Seq and transcription in Middle (**C**) and Aged (**D**) cells. Orange and dark green indicate upregulation and downregulation of Ribo-Seq, respectively, by more than twofold. Red indicates upregulation of transcription by more than twofold. The outlier is shown as a large black dot.

**Figure 5 jof-08-00938-f005:**
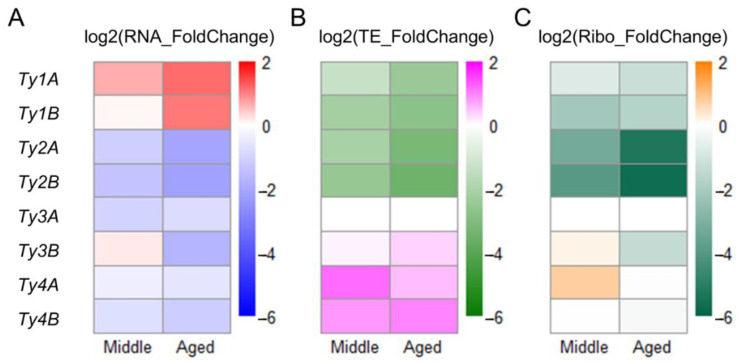
Changes in expression of *TyA* and *TyB* genes in Middle and Aged cells. Young cells were used as a control. (**A)** Changes in transposon RNA. Red and blue indicate upregulation and downregulation, respectively. (**B**) Changes in transposon translational efficiency (TE). Magenta and green indicate upregulation and downregulation, respectively. (**C**) Changes in transposon Ribo-Seq footprints. Orange and dark green indicate upregulation and downregulation, respectively. Ty5 genes were not detected.

**Figure 6 jof-08-00938-f006:**
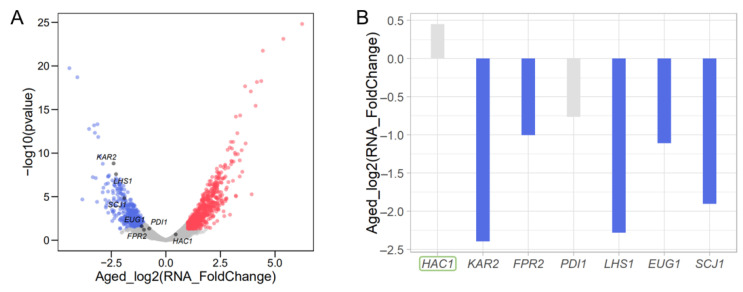
Changes in transcription levels of *HAC1* and its downstream genes. Volcano plot (**A**) and bar plot (**B**) of changes in Aged cells are shown. Red and blue indicate upregulated and downregulated genes, respectively. Young cells were used as a control. Upregulated (red) and downregulated (blue) expression changes of more than twofold compared to Young cells are shown. Grey indicates no notable changes. *HAC1* is indicated by the green box.

**Figure 7 jof-08-00938-f007:**
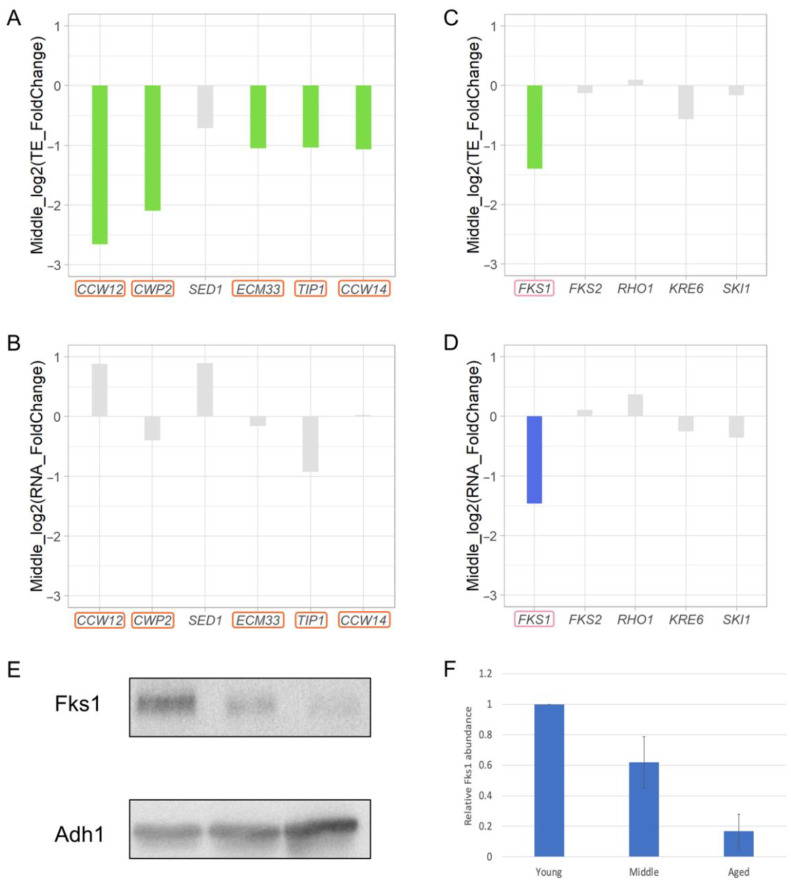
Translation of genes involved in cell wall synthesis decreases. Changes in translational efficiency (**A**) and RNA (**B**) of genes encoding major mannoprotein components and changes in translational efficiency (**C**) and RNA (**D**) of genes involved in cell wall synthesis in Middle cells are shown. Green indicates downregulation of translational efficiency by more than twofold. Blue indicates downregulation of RNA by more than twofold. Grey indicates no notable changes. Genes with decreased translational efficiency more than two-fold are indicated by the red box. Young cells were used as a control. (**E**) Western blotting analysis of Fks1 protein in Young, Middle, and Aged cells. Adh1 served as a loading control. (**F**) Relative quantitative Fks1 protein level of the Western blotting results. Young is shown as 100%. The means and standard errors of 6 experimental replicates of 2 biological replicates are shown.

**Figure 8 jof-08-00938-f008:**
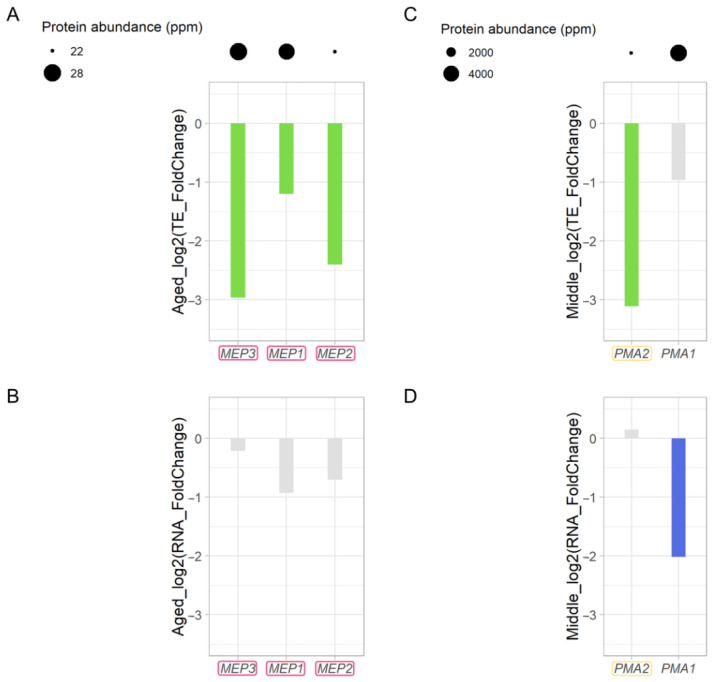
Translation of genes encoding ammonium permease and plasma membrane H^+^-ATPase decreases in Aged cells. Changes in translational efficiency (**A**) and RNA (**B**) of *MEP3*, *MEP1*, and *MEP2* genes (indicated by the red box) and changes in translational efficiency (**C**) and RNA (**D**) of *PMA2* (indicated by the yellow box) and *PMA1* genes are shown. Green indicates downregulation of translational efficiency by more than twofold. Blue indicates downregulation of RNA by more than twofold. Grey indicates no notable changes. Young cells were used as a control.

**Figure 9 jof-08-00938-f009:**
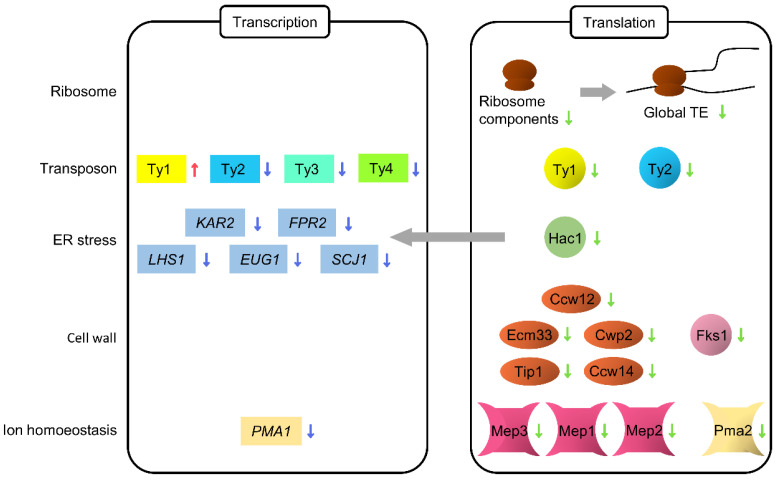
Possible transcriptional and translational regulation during replicative aging in yeast. The grey arrow shows the regulation by the transcription factor, Hac1.

## Data Availability

The ribosome profiling and RNA-Seq data used in this study were deposited in the Gene Expression Omnibus (GEO, GSE203147) of the National Center for Biotechnology Information (NCBI). Other source data are available from the corresponding author upon request.

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
