# Peer review of "Multifarious Translational Regulation during Replicative Aging in Yeast"

_jof, 2022, doi:10.3390/jof8090938_

Round 1

Reviewer 1 Report

The authors investigated translation efficiency during replicative aging and identified genes with increased and decreased translation efficiency. The manuscript is clearly written, the appropriate experimental approach is used and the results are clearly discussed.

I would suggest extending the introduction with some of the latest research on replicative aging, such as:

https://pubmed.ncbi.nlm.nih.gov/34661239/

https://pubmed.ncbi.nlm.nih.gov/32410020/

https://pubmed.ncbi.nlm.nih.gov/30650660/

https://pubmed.ncbi.nlm.nih.gov/30718408/

https://pubmed.ncbi.nlm.nih.gov/31358968/

https://pubmed.ncbi.nlm.nih.gov/34192514/

https://pubmed.ncbi.nlm.nih.gov/33045248/

https://pubmed.ncbi.nlm.nih.gov/30963997/

As for the conclusion, perhaps the authors could afford to suggest why certain genes show greater or lesser translation efficiency.

Author Response

We thank the reviewer for the positive assessment of our work.

In the revised manuscript, we referred to all the suggested papers and added sentences in the introduction (Line 55-59) and results section (Line 408-414).

We also modified the sentence in Results and Discussion, suggesting a possible mechanism of translational regulation during aging (Line 372, 392).

Reviewer 2 Report

The manuscript "Multifarious translational regulation during replicative aging

in yeast" by Zhao et al. studies the translation pathway during replicative aging of Saccharomyces cerevisiae. It gives some interesting ideas about the genes which translation is modified during aging. However, the conclusions are scarce, as it is based only in a comparation of ribosomal profiling and RNA-Seq. The conclusions are not supported with additional experiments, so the transcriptomic data is used over and over. Some conclusions are based on few examples, like the involvement of cell wall synthesis genes. The down regulation of translation is clear, but already known, and for instance, most GO categories in Fig 3A (line 259) contain translation genes, so there is not much else going on. 

Additional experiments of ammonia transport and cell wall synthesis are required. 

Author Response

(The authors gave the same response as above.)

Reviewer 3 Report

Dear authors,

 this paper clearly demonstrates that translation efficiency is increasing for some genes and is decreasing for others during aging. This is a clear difference to existent pure transcription profile papers and is therefore important.

But there is a major drawback, that has to be improved in the revised version of the manuscript. There is no single indication of biological replicates in the whole paper and statistics is completely absent.

Additionally, there are some minor remarks:

·         In line 226 the citation is missing

·         Figure 7: How was protein abundance measured?

·         the whole discussion section is wrongly designed: In the results section there is already a intensive “discussion” (what I personally like). The DNA damage response is first mentioned in 4.1. Therefore, according to the rest of the manuscript chapter 4.1 would be basically 3.10. Chapter 4.2-4.4 are in greater accuracy already discussed in the results section and could be removed . Eventually Chapter 3 should be named Results and Discussion and the pure discussion section should be discarded.

Author Response

We thank the reviewer for the valuable comments on the manuscript.

In the revised manuscript, we stated that both Young and Middle cell samples were analyzed as biological replicates (Line 116 and 120).

Aged sample was a single replicate due to cost constraints. However, DESeq2, the statistical package we used, could estimate data dispersion so thus allowing us reliable analysis. A similar approach was taken by the following papers:

https://doi.org/10.1038/s41467-021-26923-3

https://doi.org/10.7554/eLife.49117

https://doi.org/10.1016/j.cels.2020.06.011

As for statistical analysis, we calculated the adjusted p-values (or q-value) (by Benjamini and Hochberg method) through DESeq2.

To make it clearer, we added the p-value (pval) and adjusted p-value (padj) in the revised manuscript (Table S1 and Table S4).

Minor remarks:

1.We corrected Line 249.

2.We referred to the article about how to measure protein abundance (Ho et al. 2018). And, just in case, for Figure 8, the protein abundance data was from PAXdb database as described in Materials and Methods.

3.Accordingly, we rearranged the Results and Discussion sections.

Round 2

Reviewer 2 Report

No new experiments were included.

Author Response

According to the reviewer’s valuable comments, we performed Western blotting experiments and now show that the cell wall synthetase (Fks1) are in fact downregulated.

We believe that we have reached many important conclusions as we describe in “Conclusions”.

The important pathways involved in various cell functions are translationally regulated and cause diverse phenotypic changes.

Round 3

Reviewer 2 Report

The Western blot improved the manuscript

Author Response

Thanks for your positive evaluation on our manuscript.